# Synthesis, Stability, and Antidiabetic Activity Evaluation of (−)-Epigallocatechin Gallate (EGCG) Palmitate Derived from Natural Tea Polyphenols

**DOI:** 10.3390/molecules26020393

**Published:** 2021-01-13

**Authors:** Bingbing Liu, Zhengzhong Kang, Weidong Yan

**Affiliations:** 1Department of Chemistry, Zhejiang University, Hangzhou 310027, China; 11637060@zju.edu.cn (B.L.); kangzhzh@zju.edu.cn (Z.K.); 2State Key Laboratory of Environmental and Biological Analysis, Hong Kong Baptist University, Kowloon Tong, Kowloon, Hong Kong 999077, China

**Keywords:** EGCG palmitate, molecular modification, stability, antidiabetic activity, molecular docking

## Abstract

This work describes a novel approach for the synthesis of (−)-epigallocatechin gallate (EGCG) palmitate by a chemical-synthesis method, where the elevated stability of the EGCG derivative is achieved. Various parameters affecting the acylation process, such as the base, solvent, as well as the molar ratio of palmitoyl chloride, have been studied to optimize the acylation procedure. The optimized reaction condition was set as follows: EGCG/palmitoyl chloride/sodium acetate was under a molar ratio of 1:2:2, with acetone as the solvent, and the reaction temperature was 40 °C. Under the optimized condition, the yield reached 90.6%. The EGCG palmitate (PEGCG) was isolated and identified as 4′-*O*-palmitoyl EGCG. Moreover, the stability of PEGCG under different conditions was proved significantly superior to EGCG. Finally, PEGCG showed better inhibition towards α-amylase and α-glucosidase, which was 4.5 and 52 times of EGCG, respectively. Molecular docking simulations confirmed the in vitro assay results. This study set a novel and practical synthetic approach for the derivatization of EGCG, and suggest that PEGCG may act as an antidiabetic agent.

## 1. Introduction

(−)-Epigallocatechin gallate (EGCG), a compound with versatile bioactivities, is the most abundant tea polyphenol in green tea leaves [1]. EGCG has specific health-promoting activities, including antioxidative, antibacterial, anticancer, cardioprotective effect, and cholesterol-lowering effect [2,3,4,5,6]. It is the first botanical drug approved by the US Food and Drug Administration (FDA) as an anti-infective prescription for genital warts. Besides, EGCG is found to possess antidiabetic activity. Wolfram et al. found that EGCG beneficially modified glucose and lipid metabolism in H4IIE cells and markedly enhanced glucose tolerance in diabetic rodents [7]. Also, hepatic glucose production was repressed by EGCG through a PI3K-dependent manner [8]. Kamiyama et al. found that green tea supplementation increased insulin sensitivity and showed the antidiabetic effect on animal models of insulin resistance [9].

Oral administration of EGCG is found to be first absorbed in the intestine, which is a slightly alkaline condition [10]. However, EGCG is unstable in an alkaline system and is easily oxidized. For example, Sang et al. found that EGCG is unstable in sodium phosphate buffer (pH 7.4), where 90% of EGCG is lost in only 3 h [11,12]. Moreover, EGCG is highly hydrophilic, which means that it cannot easily pass through the lipid bilayer cell membranes by passive diffusion, and will be quickly inactivated or irreversibly oxidized, thus leading to the low availability [12,13]. Previous studies showed that only 0.01% of EGCG was absorbed in rats when given 56 mg of EGCG orally, and 0.32% of EGCG was absorbed in humans after a single oral intake of 97 mg [14].

Therefore, to increase the metabolic stability and bioavailability of EGCG, various efforts have been made on the structural modification of EGCG [15,16,17]. The EGCG derivatives showed enhanced bioavailability, and among these, EGCG-C16 esters, or EGCG palmitates, are found to have excellent bioactivities, such as antivirus [18,19], antitumor [20,21], antibacterial, and antifungal activities [22]. For instance, Matsumura et al. found that EGCG-C16 suppressed tumor growth in colorectal tumor-bearing mice compared to EGCG [20]. An EGCG palmitate was found to act as a topical antiviral agent for herpes simplex virus (HSV) [18]. EGCG palmitate is a novel lipophilic antioxidant that is considered to be generally recognized as safe (GRAS) by the US FDA in 2019. The enhanced bioavailability of the EGCG-acyl ester derivatives may be attributed to the slow release of EGCG from these derivatives in vivo [16].

Synthesis of EGCG palmitate can be attained by the chemical-synthesis or enzymatic method [15,19,23]. Although enzymatic catalysis is regioselective under some conditions, it has the shortcomings of being expensive, time-consuming, and limited in mild conditions due to the physical properties of the enzymes, while chemical synthesis is fast, cheap, and has higher yields; chemical synthesis includes homogeneous catalysis and heterogeneous catalysis. As for homogeneous catalysis, the advantages include that the catalytic efficiency is higher, and the catalyst usage is less compared to using the heterogeneous catalyst, while for heterogeneous catalysis, its merit is that the catalyst can be easily removed from the reaction system. Comparison of the reported chemical-synthesis or enzymatic method of EGCG palmitate with the method developed in this study is listed in Table 1. It can be clearly seen that the method developed in this study has the merit of high-yield.

As for the esterification reaction, the kinds of catalysts can be divided into acid and base. The esterification of EGCG usually uses acyl chloride or anhydride as the acylation reagent, and bases like pyridine to act as the acid-binding agent that can react with the hydrogen chloride released in the esterification process. However, bases like pyridine are toxic [15,24]. Based on these facts, we decided to select a less toxic base for the esterification of EGCG. Some solid bases like sodium carbonate have been used in the esterification process [25], therefore, we listed some potential solid bases that might be applicable for this EGCG reaction. The toxicity of these solid bases was obtained from the Material Safety Data Sheet (MSDS) database and listed in Table 2. In addition, since Mori indicated that the positions of the acyl groups of EGCG derivatives do not affect their antiviral activities [19], we determined to develop a facile chemical-synthesis method to obtain the EGCG palmitate. 

In this work, the objectives were to better understand the relationships between various reaction variables (i.e., usage of the reagent) and the responses (EGCG conversion, product yield), and to obtain the optimum synthesis condition for PEGCG. Also, the stability of the PEGCG under several conditions was studied and compared with EGCG. Finally, since many studies have reported the antidiabetic activity of EGCG and some EGCG ester derivatives [8,9,26], we deduced that PEGCG may have the same or even higher activity. The antidiabetic activity of PEGCG was evaluated and compared with EGCG as well as standard drug acarbose by using in vitro assays.

## 2. Results and Discussion

### 2.1. Screen of Different Reaction Conditions

A single-factor experiment was conducted to investigate the variables (i.e., the molar ratio of the palmitoyl chloride, base, and solvent) and the responses (conversion, product yield) of the reaction, in order to set the optimum reaction condition for the synthesis of PEGCG.

#### 2.1.1. Effect of the Molar Ratio of Palmitoyl Chloride

To investigate the effect of the molar ratio of palmitoyl chloride on the reaction, the reaction temperature (40 °C), reaction time (6 h), the quantity of EGCG (10 mM), and sodium acetate (40 mM) were held constant, and only usages of the palmitoyl chloride were changed. 

As shown in Figure 1a, the retention time (RT) of EGCG was 17.1 min, and RT of PEGCG was 44.5 min. As shown in Figure 1d, when the molar ratio of EGCG to palmitoyl chloride was 1:1, the conversion of EGCG was 94.3%. When the molar ratio was 1:2, the EGCG conversion was enhanced to 99.3%; when the molar ratio increased to 1:3 and 1:4, no EGCG peaks were observed in HPLC, and it was considered that the EGCG conversion reached nearly 100%. However, even when the palmitoyl chloride was in excess, the product was still EGCG monopalmitate (Appendix A). No esters of higher degrees (di-, tri-, or tetraesters) were formed, which might be ascribed to the steric hindrance of the long fatty acid chain that caused difficulty to the acylation process [15]. In addition, it may also be because the activity of each hydroxyl of the EGCG molecule is different. As reported before, hydroxyls on the B ring and D ring of EGCG are more reactive than other hydroxyls.

#### 2.1.2. Effect of Base

Two molar equivalents of different solid bases (sodium acetate, sodium carbonate, and sodium bicarbonate) were used, and triethylamine was used as a conventional organic base to be compared with the three selected bases. The usage of palmitoyl chloride (2 molar equivalents) and other factors were held constant.

HPLC chromatograms of the mixtures obtained with the above are shown in Figure 1b. As shown in Table 3, EGCG conversions were nearly 100%. Sodium acetate and triethylamine gave a high yield of PEGCG, which was 90.6% and 88.1%, respectively, while it was low with sodium acetate (57.1%) and sodium carbonate (72.4%). Therefore, sodium acetate was considered as a suitable base and was used in the following experiments.

#### 2.1.3. Effect of Solvent

The 1-octanol/water partition coefficient, log P, is recognized as one of the principal parameters to evaluate the lipophilicity of chemical compounds. The compound of higher lipophilicity has a higher log P value and is considered relatively less polar. The lipophilicity of the EGCG, 4′-*O*-palmitoyl EGCG and different solvents were computationally obtained by ALOGPS 2.1 (http://www.vcclab.org/lab/alogps/) [27].

The HPLC chromatograms of the products of different solvents are shown in Figure 1c. Under the conditions of 2 molar ratio of palmitoyl chloride and sodium acetate as the base, EGCG conversions were almost the same by using different solvents, and the products were mono-esters. However, as shown in Table 4, the yield of the product was not proportional to the polarity of the solvent; acetone showed the highest yield (90.6%) and other solvents showed relatively lower yields (no more than 90.6%). Dimethylformamide is a relatively polar solvent, while ethyl acetate and diethyl ether are relatively non-polar solvents. Since the log P value of the material (EGCG) is 2.38 and the product (PEGCG) is 6.48, it might be speculated that acetone acts as a medium-polarity solvent that offers a compromised dissolution condition between the materials and the product. It had been found that 10 mmol EGCG was partly dissolved in ethyl acetate and diethyl ether, respectively, which might explain the low yield with the corresponding solvent. 

Based on these results, acetone was considered the most desirable solvent.

In conclusion, considering the reaction variables on the effect of the reaction, the optimum reaction condition was set as follows: sodium acetate as the base, acetone as the solvent, and the molar ratio of palmitoyl chloride was 2. Under the optimum reaction condition, the yield of EGCG palmitate could reach 90.6%.

### 2.2. Stability under the Alkalescent Condition

According to previous studies, EGCG was extremely vulnerable to degrading in alkaline solutions. If EGCG were not well absorbed, part of the mechanism might include the selective degradation of EGCG in the intestine where the pH is neutral or alkaline. The oxidation products of EGCG are two EGCG dimers: P2 (Mw 884) and Theasinensin (Mw 914) [11,28].

Firstly, the oxidation products of EGCG in ethanol/PBS solution were identified as P2 (RT 23.199 min) and Theasinensin (RT 23.199 min) by HPLC-MS (Appendix A).

Then, the stability of EGCG and PEGCG under the alkalescent condition was compared. As shown in Figure 2a,b, under the alkaline condition in the open air, EGCG immediately turned into its dimers once added into the solution. After 1 h, 93% of the EGCG was degraded. After 5 h, the EGCG was utterly degraded. While for PEGCG, most of the PEGCG was hydrolyzed to EGCG and its dimers after 1 h. After 5 h, 100% of the PEGCG was hydrolyzed to EGCG and then oxidized to EGCG dimers. The above results indicate that the PEGCG is significantly more stable than EGCG under the alkalescent condition. 

### 2.3. Storage Stability

After 60 days in the open air, the white-colored EGCG powder turned to reddish-brown upon long-term storage, while the color of the PEGCG remained nearly the same. It showed that 78% of EGCG palmitate remained after being stored for 60 d at room temperature (Figure 2c), while EGCG was 45% remained (Figure 2d). This indicates that PEGCG keeps better stability in the air, making it is more convenient for storage.

### 2.4. Thermal Stability

As shown in Appendix A, the thermal decomposition temperature of EGCG was 234.5 °C, while the thermal decomposition temperature of PEGCG was 311.19 °C. PEGCG had better thermal stability compared with EGCG, which is vital since PEGCG can be used as an antioxidant in edible oil. 

Edible oil is usually used at a temperature under 250 °C. At this time, PEGCG, a lipid-soluble antioxidant, can still keep stable and play the antioxidant role at high temperatures, protecting the nutritional components of edible oil from being destroyed. What is more, acrylamide and pyridine heterocyclic amine (PHIP), the carcinogenic and mutagenic compounds, are quickly produced when foods are frying at high temperatures. The inhibitory effects of polyphenols on acrylamide and PHIP have been reported [29,30]. Given the excellent thermal stability and antioxidant property of PEGCG palmitate, it is speculated that PEGCG could also inhibit the production of acrylamide and PHIP, which deserves further study.

### 2.5. Inhibition of α-Amylase

According to various in vivo studies, inhibition of α-amylase and α-glucosidase is assumed to be one of the most effective approaches for diabetes care [31]. However, several commercial antidiabetic drugs, such as acarbose, miglitol, voglibose, and sitagliptin, have positive effects on glycemic values after food intake, but their lack of specificity has produced some gastrointestinal side effects, like abdominal cramping, flatulence, and diarrhea [32,33]. Natural α-glucosidase and α-amylase inhibitors are therefore being investigated as new candidates to control hyperglycemia in diabetic patients, for they do not cause severe side effects and may also be beneficial in weight reduction for people consuming large amounts of starch [34]. 

For α-amylase inhibition, as shown in Table 5, the inhibitory activity in descending order was acarbose (IC_50_ 1.10 μM) > PEGCG (IC_50_ 1.64 μM) > EGCG (IC_50_ 7.44 μM) (Figure 3). It is notable that the inhibitory activity of PEGCG was 4.5 times higher than EGCG, and was comparable to acarbose. The elevated inhibitory activity of PEGCG may be attributed to its increased stability and bioavailability.

### 2.6. Inhibition of α-Glucosidase

Similarly, the inhibition of PEGCG on α-glucosidase was significantly improved compared to EGCG. It was found that IC_50_ values of EGCG, PEGCG, and acarbose were 11.50, 0.22, and 0.15 µM, respectively. Thus, the α-glucosidase inhibitory activity of PEGCG was 52 times higher than EGCG and 0.68 times of acarbose. This result indicated that the PEGCG strongly suppressed the α-glucosidase activity and has the possibility of controlling postprandial hyperglycemia.

### 2.7. Molecular Docking of the Ligands with α-Amylase

Here we used the molecular docking method to obtain the possible binding conformation of drugs and enzyme. Ten most likely binding structures are produced from AutoDock software. We compare the binding energy, contact atom number, and contact surface of the ten conformations (Appendix A). The top 3 binding conformations are listed in Figure 4 and Figure 5, while other possible binding structures are shown in Appendix A.

The catalytic sites of amylase are ASP231, GLU261, and ASP328, which are located at the bottom of the active pocket. As shown in Figure 4, drug molecules are all in the pocket and binding directly to the catalytic sites. It indicates that the possible inhibition mechanism is achieved by occupying the catalytic sites to prevent the binding of substrates (sugar). It can be seen from Figure 4 that the binding conformations of the three drugs are different. Acarbose covered more catalytic sites than the other two drugs, which may be the reason why acarbose had better inhibitory effect. The three aromatic rings of EGCG could not completely cover the catalytic sites due to the spatial structure. More interestingly, the tail chain of PEGCG could not only anchor on the enzyme to assist binding but also bind to the catalytic sites (as shown in Figure 4, binding model 1). This may be the reason why PEGCG has a better inhibition effect than EGCG. 

Due to the limitation of AutoDock software, we could not get the accurate binding free energy of ligand-enzyme. We simply used the number of amino acids in contact and hydrogen bonds as the representative of van der Waals interaction and electrostatic interaction, respectively. The number of contact residues and number of the hydrogen bonds are listed in Table 6 (Binding details are shown in Appendix A). As shown in Table 7, acarbose has the maximum number of hydrogen bond donor and acceptor counts. For PEGCG, the palmitoyl structure does not increase the number of hydrogen bond donors and acceptors. 

### 2.8. Molecular Docking of the Ligands with α-Glucosidase

Different from amylase, the catalytic sites ASP215, GLU277, and ASP352 of glucosidase are located below the pocket, and the substrate needs to pass through the pocket before binding to the catalytic site.

The three ligand molecules are all inserted into the catalytic pocket in Figure 5, without direct contact with the catalytic site. Therefore, the possible inhibition of the ligand on glucosidase is achieved by blocking the pocket that leads to the catalytic site. It can be seen from the structure figure that the aromatic structures of EGCG and acarbose are deeply inserted into the pocket and stay close to the catalytic sites, which successfully blocks the possible binding of substrate. As for PEGCG, both aromatic structure and palmitoyl structure can enter the pocket. As shown in binding model 1 (Figure 5), the palmitoyl part with suitable length can be folded into the pocket or anchor on the edge of the pocket to assist binding. At the same time, the hydrogen bond number is displayed as EGCG (3) < PEGCG (4) < acarbose (5), and the number of amino acids in contact is EGCG (19) < acarbose (26) < PEGCG (27) in Table 8 (Binding details are shown in Appendix A). Interestingly, the PEGCG is much closer to acarbose in terms of amino acid numbers in contact and hydron bond numbers. 

We also observed that the number of contact atoms and the contact area of PEGCG were also close to that of acarbose and larger than EGCG (Appendix A). This indicates that although the mother nuclei structures of PEGCG and acarbose are different, they show similar characteristics in binding with glucosidase. This similarity was also found in the IC50 data of PEGCG and acarbose. From the binding conformations in Figure 5, we found that in addition to blocking the binding pocket, some part of PEGCG and acarbose are anchored at the edge of the pocket, while EGCG could only stay inside the pocket. Combined with IC50 data, we speculate that glucosidase may prefer inhibitors with a long chain, which not only block the inside of the pocket, but also cover the opening of the pocket.

## 3. Experimental Section

### 3.1. Materials

EGCG (from green tea, purity 92%) was supplied by Pulimeidi Inc. (Hangzhou, China). Palmitoyl chloride, α-amylase (from rhizobacteria *Bacillus Licheniformis*, EC 3.2.1.1), and acetonitrile of HPLC grade were purchased from Sigma-Aldrich Co. (St. Louis, MO, USA). α-Glucosidase (from yeast *Saccharomyces Cerevisiae*, EC 3.2.1.20, 18.5 unit/mg) was purchased from Sinopharm Group Chemical Reagent Co. (Shanghai, China). Sodium acetate, sodium carbonate, sodium bicarbonate, and triethylamine were from Sinopharm Group Chemical Reagent Co., Ltd. (Shanghai, China). All of the other solvents and chemicals used were obtained from Sinopharm Group Chemical Reagent Co., Ltd. (Shanghai, China) and were of analytical grade. Sodium phosphate buffer (PBS) was purchased from Solarbio Co. (Beijing, China) and was prepared to make a concentration of 0.01 M aqueous solution (pH 7.2).

### 3.2. Synthesis Procedure

In a typical one-step synthesis protocol, 10 mmol EGCG (4.98 g) was added to 100 mL of a solvent in a 250 mL flask immersed in a water bath (40 °C). After complete dissolution, the base was added to the solution. Subsequently, different molar ratios of palmitoyl chloride (1, 2, 3, or 4 molar ratio) were dropwise added to the solution, respectively, with mechanical stirring (100 rpm). After 6 h, the reaction mixture was filtered and washed with 100 mL deionized water. Then 100 mL ethyl acetate was added for the extraction of the acylation product. And the organic phase was washed 2 times with deionized water. After that, the upper organic phase was dried with anhydrous sodium sulfate, concentrated under reduced pressure, and lyophilized. The lyophilized products were preserved at −20 °C for further use.

### 3.3. HPLC-MS Analysis

The composition of the acylation mixture under the optimum reaction condition was determined by HPLC-MS, which used an Agilent 1290 HPLC unit (Agilent Technologies, Palo Alto, CA, USA) with a UV detector and a Cosmosil ODS C18 column (4.6 mm × 250 mm, 5 µm; Nacilai Tesque Inc., Kyoto, Japan). Eluent A and eluent B were acetonitrile/water = 10:90 (*v*/*v*), and acetonitrile/water = 80:20 (*v*/*v*), each containing 0.2% formic acid (*v*/*v*). A gradient elution program was as follows: 0–20 min, linear gradient 0–10% B; 21–60 min, 88% B isocratic. The flow rate was 1.0 mL/min, and fractions were detected at 275 nm. Mass spectroscopic (MS) analysis was performed using a 6460 Triple Quad MS detector system (Agilent Technologies, Palo Alto, CA, USA). The eluent was introduced into an electrospray source at negative mode (desolvation temperature 325 °C, capillary voltage 3.5 kV, nebulizer 45 psi). Argon was used as the collision gas (collision energy 16 eV), and nitrogen (dry gas flow, 5 L/min) was the desolvation gas. 

### 3.4. Purification and Identification of the Acylation Product

Flash column chromatography was used to separate individual EGCG palmitate. 10 g of EGCG derivatives were eluted on a silica column with a mixture of petroleum ether/ethyl acetate/acetic acid (3:1:0.05, *v*/*v*/*v*). The fractions were monitored by TLC (petroleum ether/ethyl acetate/acetic acid = 1:1:0.05, *v*/*v*/*v*), and the predominant fraction was collected and washed three times with deionized water. Solvents were removed using a rotary evaporator. The predominant fraction (compound 1) was analyzed by NMR for its specific structure. The melting point of compound 1 was obtained by the differential scanning calorimetry (DSC) analysis on a DSC instrument (Q100, TA Instruments, New Castle, DE, USA).

The crude products were separated into different fractions by silica column chromatography, and the predominant one was collected and identified by NMR, which was reported in our previous study [35].

Theoretically, the molecular weight of EGCG monopalmitate is 696.8, and EGCG is 458.4 Da. As shown in Figure 6b, the selective ion of m/z 695 indicated that peaks 2, 3, and 4 might be EGCG monopalmitate. However, only the MS spectrum of peak 2 showed m/z 695.3 ([M-H]^−^) and m/z 238 ([M-H-palmitoyl]^−^), which is the indication of EGCG monopalmitate (PEGCG). Peak 3 and 4 were identified as the byproducts (Figure 6d and Appendix A). According to Appendix A, mass spectra of peak 5 and 6 showed molecular ion peak of m/z 737, indicating that the components were not EGCG monopalmitate, either.

The ^1^H and ^13^C NMR analyses of PEGCG were as follows (Figures in Appendix A): PEGCG: yield 90.6%; white crystal; m.p. 96.7 °C (Appendix A); ^1^H NMR (400 MHz, DMSO-*d*_6_). δ 9.17 (br, OH), 6.81 (s, 2H, H-2″, H-6″), 6.41 (s, 2H, H-2′, H-6′), 5.93 (d, *J* = 2.2 Hz, 1H, H-8), 5.82 (d, *J* = 2.2 Hz, 1H, H-6), 5.33 (s, 1H, H-3), 4.94 (s, 1H, H-2), 2.90 (m, 1H, H-4a), 2.65 (m, *J* = 16.0 Hz, 1H, H-4b), 2.27 (t, *J* = 7.4 Hz, 2H, H-p-2), 1.50 (m, 2H, H-p-2), 1.09–1.23 (m, *J* = 53.6 Hz, 24H, H-p-4 to 15), 0.85 (t, *J* = 6.8 Hz, 3H, H-p-16), 4.09 (br, OHs).

^13^C NMR (101 MHz, DMSO-*d*_6_). δ 173.44 (C-p-1), 165.31 (C-11), 156.58 (C-7), 156.53 (C-9), 155.66 (C-5), 145.69 (C-3″, C-5″), 145.48 (C-3′, C-5′), 138.83 (C-4″), 132.40 (C-4′), 128.65 (C-1′), 119.10 (C-1″), 108.67 (C-2″,C-6″), 105.51 (C-2′, C-6′), 99.59 (C-10), 97.38 (C-8), 94.35 (C-6), 76.54 (C-2), 68.08 (C-3), 33.31 (C-p-2), 31.36 (C-p-3), 28.50-29.09 (C-p-4 to 13), 25.76 (C-4), 24.48 (C-p-4), 22.16 (C-p-15), 14.01 (C-p-16). P, palmitoyl. 

Based on the HPLC-MS and NMR analyses, PEGCG was identified as 4′-*O*-palmitoyl EGCG (Figure 3, molecular structures coincide with PubChem), where acylation occurs on the B ring of the EGCG molecule. As reported by Chen et al. [24], the same 4′-*O*-palmitoyl EGCG was synthesized and identified. It has been reported that hydroxyls on the B ring of EGCG are more reactive [36,37,38], where the oxidation and acylation first occur. 

### 3.5. HPLC Analysis of the Acylation Reaction Process

To further analyze the mixtures of different reaction conditions (base, solvent, molar ratio of palmitoyl chloride), the lyophilized crude products were dissolved with acetonitrile and filtered respectively. 20 µL of the above solution was subjected to HPLC analysis (Wufeng LC-100 HPLC-UV, Shanghai, China) and separated on a Cosmosil ODS C18 column (4.6 mm × 250 mm, 5 µm; Nacilai Tesque Inc., Kyoto, Japan). The detection wavelength was set at 275 nm, and the column temperature was 30 °C. The flow rate was 1.0 mL/min. The elution program was the same as HPLC-MS as described above.

The content of EGCG or PEGCG in each sample was calculated according to its standard curve, respectively. Thus, the conversion of EGCG can be obtained by the following equation:
(1)$$\%\;\mathrm{Conversion}\;=\left(1-content_{\mathrm{EGCG}}\right)\times 100$$

The product’s yield was expressed as the weight of esters obtained/weight of esters (g/g) expected if EGCG is fully reacted.

### 3.6. Evaluation of pH Stability in Alkalescent Condition

Since the intestine system is alkalescent, it is necessary to assess the stability of EGCG and PEGCG under this condition. Ethanol/PBS = 80:20 (*v*/*v*, pH 7.2) was used to offer the alkalescent condition, since PEGCG was insoluble in PBS.

Equivalent molar concentrations (1.3 mM) of EGCG and PEGCG were incubated in Pyrex test tubes (20 × 1.6 cm, i.d.) in the open air at 37 °C, respectively, without any agitation. Aliquots (20 µL) of the incubation solution was periodically (1, 2, 5 h) sampled and subjected to HPLC analysis, respectively. HPLC condition was the same as Item 2.5.

### 3.7. Evaluation of Storage Stability

To study the storage stability of EGCG and PEGCG powder, 1 molar of the two powders were kept in the open air at room temperature, respectively. After 60 days, aliquots of the two powders were sampled and subjected to HPLC analysis, respectively. The HPLC condition was the same as Item 2.5.

### 3.8. Evaluation of Thermal Stability

The thermal stability behaviors of EGCG and PEGCG were investigated on a TG apparatus (SDT Q600, TA Instruments, New Castle, DE, USA). About 3.0 mg of each sample was used to perform the analysis by putting the powder on a TG pan. The sample was scanned from 30 to 400 °C with a heating rate of 10 °C/min under a nitrogen atmosphere.

### 3.9. α-Amylase Inhibition Assay

The α-amylase activity was measured using the method described by Nampoothiri with slight modifications [39]. For subsequent assays, equal volumes (100 μL) of sample (EGCG and acarbose were dissolved in aqueous solution, and PEGCG in ethanol) and 100 μL of porcine pancreatic α-amylase (0.5 mg/mL) were incubated in microtubes at 25 °C for 10 min. After pre-incubation, a volume of 1% starch solution in 10 mM sodium phosphate buffer (pH 7.2) was added to each tube and the samples were incubated at 25 °C for a further 10 min. The reaction was stopped with 200 μL of dinitrosalicyclic acid color reagent and tubes were incubated in boiling water for 5 min. Once the samples were cooled to room temperature, 50 μL was removed from each tube and transferred to the wells of 96-well microplate. The reaction mixture was diluted by adding 200 μL of water to each well, and absorbance was measured at 540 nm. Blank readings (no enzyme) were subtracted from each well, and results were compared to the control. The pharmacological inhibitor, acarbose, was included as a positive control. Two controls were used, which is, control 1 for EGCG and acarbose (20 μL of PBS was added instead of the sample), and control 2 for PEGCG (20 μL of ethanol was added instead of the sample). The inhibition of α-amylase was calculated as follows:(2)$$\%\;\mathrm{Inhibition}=\left(1-\frac{A_{\mathrm{sample}}}{A_{\mathrm{control}}}\right)\times 100$$

### 3.10. α-Glucosidase Inhibition Assay

The activity of α-glucosidase was determined in a 96-well plate based on 4-Nitrophenyl beta-d-glucopyranoside (PNPG) as a substrate. A mixture of 100 μL of enzyme solution (1 U/mL α-glucosidase in 10 mM sodium phosphate buffer) and 50 μL of the specified concentration of samples (EGCG and acarbose dissolved in aqueous solution, and PEGCG in ethanol) was pre-incubated at 37 °C for 10 min; then 50 μL of substrate solution (5 mM PNPG in 0.1 M sodium phosphate buffer) was added to each well and incubated for 15 min at 37 °C and stopped by adding 80 μL of 0.2 M Na_2_CO_3_ [39]. The amount of PNP released was measured at 405 nm. The controls (that do not include the enzyme) were prepared by adding PBS instead of the inhibitor solutions. Two controls were used, which were control 1 for EGCG and acarbose (20 μL of PBS was added instead of the sample), and control 2 for PEGCG (20 μL of ethanol was added instead of the sample). The inhibition of α-glucosidase was calculated as follows:(3)$$\%\;\mathrm{Inhibition}=\left(1-\frac{A_{\mathrm{sample}}}{A_{\mathrm{control}}}\right)\times 100$$

### 3.11. Molecular Docking Simulation

Molecular docking was used to study the binding details of the ligand-receptor complexes. AutoDock (version 4) [40,41] with Lamarckian genetic algorithm were used in the molecular docking experiments. The 3D crystal structures of α-amylase (PDB ID: 1BLI) [42] and α-glucosidase (PDB ID: 3A4A) (https://doi.org/10.1111/j.1742-4658.2010.07810.x) were obtained from protein databank (RCSB PDB). The ligands were EGCG, PEGCG, and acarbose, and the 3D structure of the ligand was built by ChemBio3D Ultra 14.0. All possible binding models obtained from AutoDock were displayed to represent the binding conformation of drugs and enzyme. The binding models are clustered by binding energy from AutoDock with RMSD of 0.2 nm. The contact region is defined as the protein within 0.5 nm of drugs. And the number of contact atom and amino acids are calculated. The contact surface area was calculated from the solution’s accessible surface area. Binding conformations are displayed by VMD [43].

## 4. Conclusions

The results of this study suggest the possibility of a novel and efficient synthesis method of lipophilic EGCG palmitate. In this study, the usage of palmitoyl chloride, base, and solvent on the effect of the product yield and EGCG conversion were elucidated. Furthermore, the stability in alkalescent condition, storage stability, as well as the thermal stability of EGCG and PEGCG were analyzed and compared with each other. PEGCG was significantly more stable than EGCG, which contributed to its enhanced bioavailability and storage convenience. Moreover, the EGCG palmitate is capable of inhibiting α-amylase and α-glucosidase, indicating its potential as an antidiabetic prodrug. Furthermore, the oral toxicity tests of EGCG palmitate needs to be further studied.

## Figures and Tables

**Figure 1 molecules-26-00393-f001:**
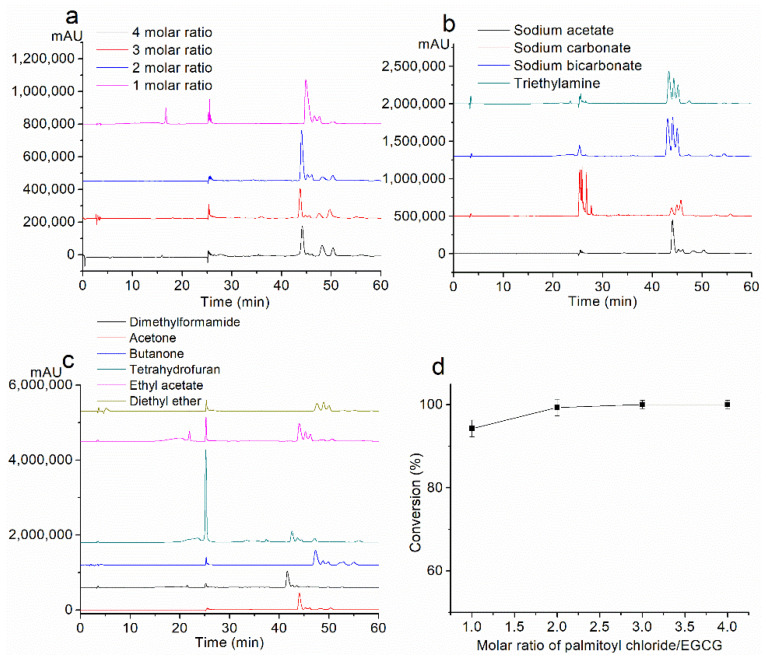
Effect of (**a**) molar ratio of palmitoyl chloride, (**b**) base, and (**c**) solvent, on the effect of the reaction by HPLC analysis. (**d**) EGCG conversion under the condition of 1–4 molar ratios of palmitoyl chloride.

**Figure 2 molecules-26-00393-f002:**
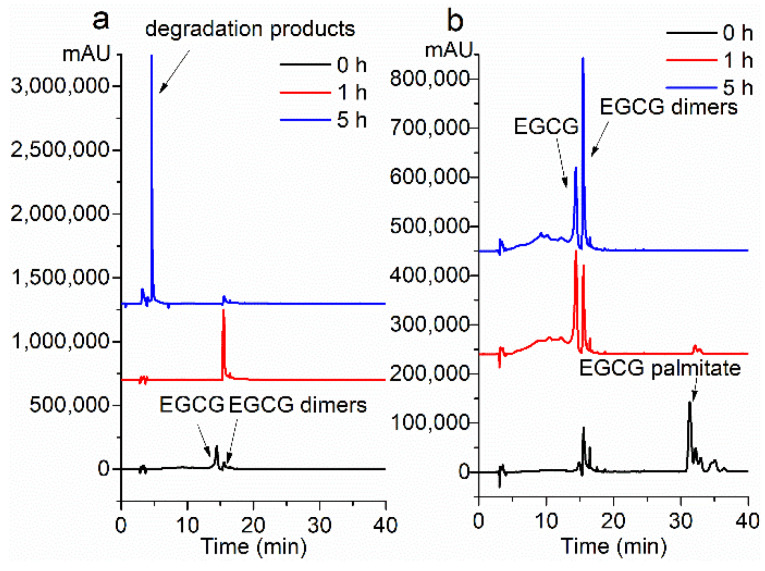
Stability evaluation under the alkalescent condition of (**a**) EGCG, and (**b**) PEGCG by HPLC-MS analysis. Storage stability evaluation of (**c**) EGCG, and (**d**) PEGCG by HPLC-MS analysis.

**Figure 3 molecules-26-00393-f003:**
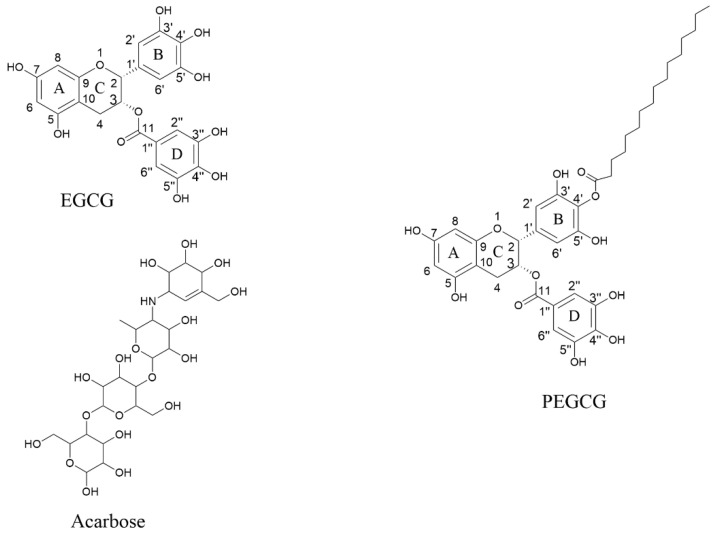
Chemical structure of the ligands from PubChem EGCG, PEGCG, and acarbose.

**Figure 4 molecules-26-00393-f004:**
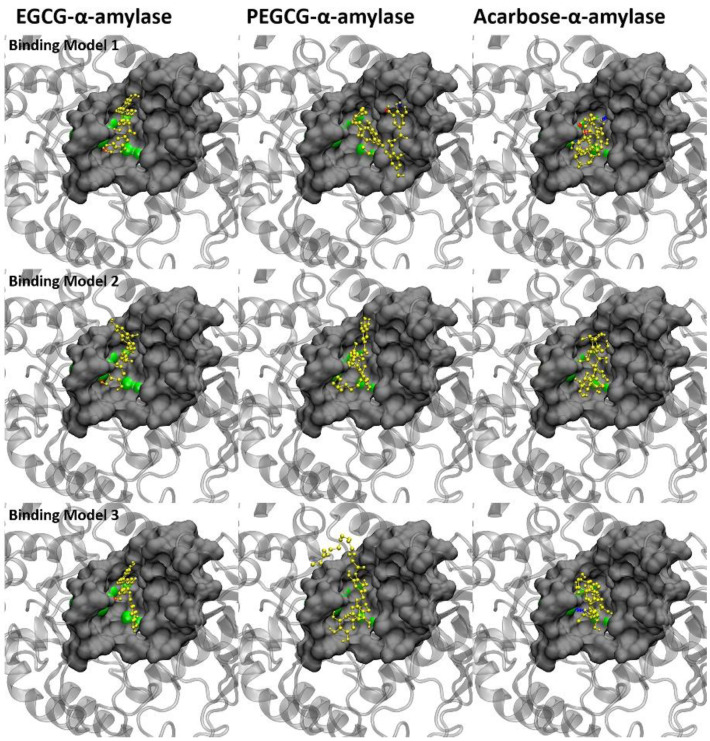
Top 3 predicted binding models of the ligands with α-amylase. Protein is shown in silver in the cartoon model. Drugs are shown in yellow. The catalytic sites are displayed in green. The ligand molecules are represented in yellow. Amino acids in contact are displayed in the surface model. The red and blue dotted lines stand for hydrogen bonds.

**Figure 5 molecules-26-00393-f005:**
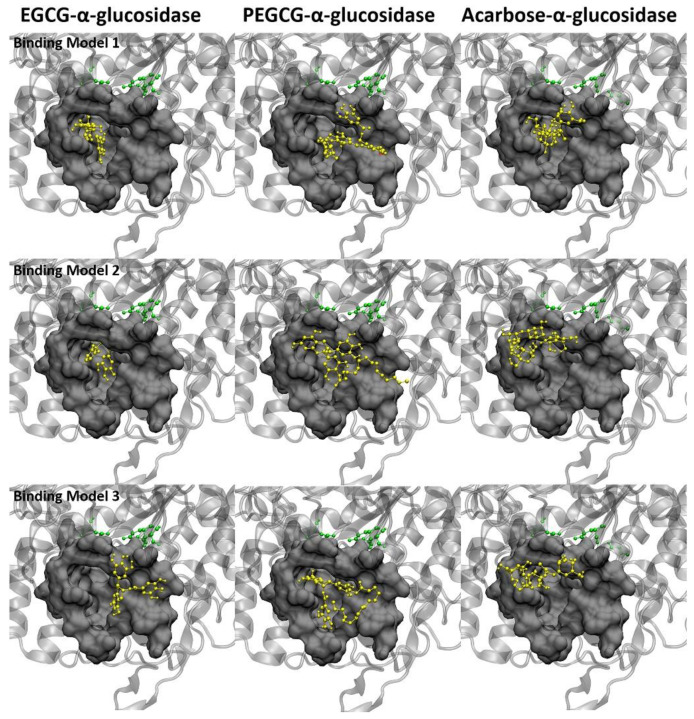
Top 3 predicted binding models of the ligands with α-glucosidase. Protein is shown in silver in the cartoon model. Drugs are shown in yellow. The catalytic sites are displayed in green. The ligand molecules are represented in yellow. Amino acids in contact are displayed in the surface model. The red and blue dotted lines stand for hydrogen bonds.

**Figure 6 molecules-26-00393-f006:**
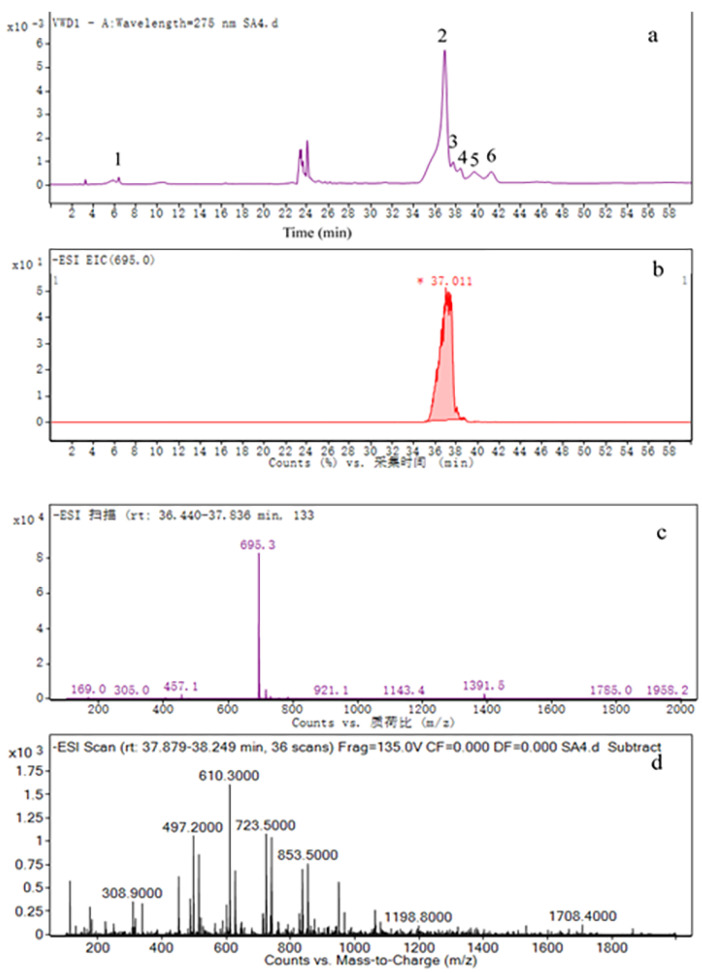
HPLC-MS analysis of product under the optimum condition (EGCG/palmitoyl chloride/sodium acetate = 1:2:2), (**a**) chromatogram of the product; (**b**) selective ion chromatogram of PEGCG (m/z 695); (**c**) mass spectra of PEGCG (Mw 696.8); (**d**) mass spectra of peak 3.

**Table 1 molecules-26-00393-t001:** Comparison of chemical-synthesis or enzymatic method of EGCG palmitate.

Method	Product	Catalyst	Solvent	Reaction Time	Yield	Ref.
Enzymatic method	Mixture of four EGCG-C16 regioisomers	Lipase PL (Alcaligenes sp.)	N,N-Dimethylformamide	8 h	35–39%	[19]
Chemical-synthesis	Mixture of four EGCG-C16 regioisomers	Triethylamine	Tetrahydrofuran	24 h	12–16%	[19]
Chemical-synthesis	EGCG-4′-*O*-palmitate	Pyridine	Ethyl acetate	3 h	−	[24]
Chemical-synthesis	EGCG-C16 tetraester	Pyridine	Ethyl acetate	−	57%	[15]
This method	EGCG-4′-*O*-palmitate	Sodium acetate	Acetone	6 h	90.6%	−

**Table 2 molecules-26-00393-t002:** Toxicity of some reagents and their harm to human health.

Reagent	Acute Toxicity Test (Mouse) ^a^	Harm to Human Health ^a^
Sodium acetate	LD50 25,956 mg/kg, non-toxic	None
Sodium carbonate	LD50 4090 mg/kg, mild toxicity	Irritant and corrosive
Sodium bicarbonate	LD50 4220 mg/kg, mild toxicity	Slight irritation to the eyes
Triethylamine	LD50 460 mg/kg, medium toxicity	strong irritant to the respiratory tract; after inhaling can cause pulmonary edema or even death
Pyridine	LD50 1580 mg/kg, mild toxicity	Strong irritant; anesthetizes the central nervous system
4-dimethylaminopyridine (DMAP)	LD50 250 mg/kg, medium toxicity	Irritating to the eyes; eating poisonous

^a^ The Acute toxicity and harm to human health were obtained from Material Safety Data Sheet (MSDS).

**Table 3 molecules-26-00393-t003:** Effect of the base on the acylation reaction.

Base	EGCG Conversion/%	Yield/% ^a^
Sodium acetate	nearly 100	90.6 ± 2
Sodium carbonate	nearly 100	57.1 ± 2
Sodium bicarbonate	nearly 100	72.4 ± 2
Triethylamine	nearly 100	88.1 ± 2

^a^ The results are calculated as mean ± standard deviation of three replicates.

**Table 4 molecules-26-00393-t004:** Effect of solvent on the acylation reaction.

Solvent	Log P	EGCG Conversion/% ^a^	Yield/% ^a^
Dimethylformamide	−0.77	95.6 ± 2	45.3 ± 3
Acetone	−0.29	nearly 100	90.6 ± 2
Butanone	0.41	nearly 100	66.2 ± 3
Tetrahydrofuran	0.35	92.5 ± 1	39.4 ± 3
Ethyl acetate	0.74	92.3 ± 1	79.3 ± 2
Diethyl ether	1.12	nearly 100	47.6 ± 2

^a^ The results were calculated as mean ± standard deviation of three replicates.

**Table 5 molecules-26-00393-t005:** Inhibition of α-amylase and α-glucosidase.

Compound	IC_50_ (µM) for α-Amylase	IC_50_ (µM) for α-Glucosidase
EGCG	7.44 ± 0.05	11.50 ± 0.02
PEGCG	1.64 ± 0.04	0.22 ± 0.03
Acarbose	1.10 ± 0.03	0.15 ± 0.02

**Table 6 molecules-26-00393-t006:** Binding overview of the ligands with α-amylase.

α-Amylase	EGCG	PEGCG	Acarbose
Number of contact residues	20	22	19
Number of the hydrogen bonds	4	2	4

**Table 7 molecules-26-00393-t007:** Hydrogen bond donor and acceptor count of the ligand.

Inhibitor	Hydrogen Bond Donor Count	Hydrogen Bond Acceptor Count
EGCG	8	11
PEGCG	7	12
Acarbose	14	19

**Table 8 molecules-26-00393-t008:** Binding overview of the ligands with α-glucosidase.

α-Glucosidase	EGCG	PEGCG	Acarbose
Number of contact residues	19	27	26
Number of the hydrogen bonds	3	4	5

## Data Availability

The data presented in this study are available on request from the corresponding author.

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
