# Peer review of "Synthesis, Stability, and Antidiabetic Activity Evaluation of (−)-Epigallocatechin Gallate (EGCG) Palmitate Derived from Natural Tea Polyphenols"

_molecules, 2021, doi:10.3390/molecules26020393_

Round 1
Reviewer 1 Report
This manuscript describes the development of synthetic approach for the derivatization of (−)-epigallocatechin gallate (EGCG) to EGCG-palmitate (PEGCG), and its activity inhibitory effects on α-amylase and α-glucosidase. Additionally, PEGCG showed pH stability in alkalescent condition, and had thermal and long term storage stability. The authors concluded that PEGCG could be used as an antidiabetic agent. This is an interesting study, the data is well presented, and the manuscript is organized logically. There are just some minor issues to be answered: 1. How the log P of EGCG and PEGCG were obtained? Please specify. 2. Please show the reaction mechanisms of synthesizing PEGCG.Author Response
Please see the attachment.

Reviewer 2 Report
Although the aim of the study was to optimize and understand the reaction for synthesis of EGCG, the introduction mainly describes the biological effect of EGCG. THE introduction can be improved highlighting the previous efforts to optimize its production.
How did you draw the synthesis of EGCG, please provide references
Reviewer 3 Report
The manuscript of Liu et al. describes a novel synthesis of (-)-epigallocatechin gallate palmitate (PEGCG) and the study of its potential antidiabetic activity by enzyme inhibition assays (α-amilase and α-glucosidase).
The paper is quite interesting. However there are some issues.
First of all, the title must be changed, in fact it seems that the PEGCG derives from natural sources, while the starting product (EGCG) is present in tea.
The use of mL, μL instead of mL or μl is recommended.
Enter as a note in table 1 where the listed acute toxicities are derived.
Line 134-137. Acetone is a medium polarity solvent and can help homogenize the reaction environment, but the product (PEGCG) may be insoluble (it favors the reaction and makes purification simple for filtration). Why is a compromised dissolution of the product also necessary?
Line 210: not "the enzyme includes hydrogen bond > van der Waals force" but "the enzyme includes hydrogen bond and van der Waals force". The magnitude of the interaction force are expressed in the next sentence.
Line 241-243. Not "Table 7, The number" but "Table 7, the number". Furthermore, the order of the contact residues number is incorrect: not "EGCG (16) < acarbose (23) < PEGCG (21)" but "EGCG (16) <PEGCG (21) <acarbose (23)".
All 7 OH signals are missing in the 1H-NMR data. Is there any explanation? DMSO-d6 is typically free of water.
Line 346. Not "PEGCG palmitate" but "PEGCG"
Line 378-381. It is advisable to delete the sentences. Paragraph 3.10 can start from line 382 with the direct description of the bioassay (like step 3.9).
Provided that these issues are settled the paper is worthy of pubblication.
Author Response
Please see the attachment.

This manuscript is a resubmission of an earlier submission. The following is a list of the peer review reports and author responses from that submission.
Round 1
Reviewer 1 Report
The present study describes the synthesis of EGCG palmitate derived from natural tea polyphenols. Stability and inhition studies have also been performed. Although the study is very interesting, the manuscript requires a deep revision to be published.
Title: What is EGCG? The authors should avoid here such abbreviation.
Abstract: The authors should describe the optimal experimental conditions and maximum yield.
Lines 26-27, 46-47, and 57-58: Please, add references.
Lines 57-64: The authors should better explain the potential chemical catalysts used for such a purpose (homogeneous or heterogeneous catalysts). Among them, solid acid catalysts are more suitable from an industrial point of view due to easy recovery and reuse, as well as low wastes production during separation and purification steps. A deep revision in this topic is required. It is more interesting to the readers.
Lines 66-70: How is this system different to other reports to merit publication? Has been a similar process reported in a previous study?? Please, report.
Line 73: Please, report EGCG source (plant specie) used in this study. Information about the catalyst used in this set of experiments is strongly required.
Lines 74 and 76: Please, correct cientific nomenclatures of enzyme sources (Bacillus licheniformis and Saccharomyces cerevisiae).
Lines 81-91: The authors should better describe the general ester synthesis procedure (temperature control – immersed reaction in a water bath or jacketed reactor; magnetic or mechanical stirring and frequency; stirrer conformation). Please, describe. Again, information about the catalyst used in this study is strongly required.
Lines 131-133: The authors should better explain what is PBS (phosphate buffer solution??). In addition, SODIUM or POTASSIUM? Ionis strenght used in this set of experiments.
Lines 134-138: Explain how the temperature control has been performed in this set of experiments.
Line 139 and others: I suggest replace in all text “long-term stability” by “storage stability”.
Line 153: “10 mM sodium phosphate buffer (pH 6.9)”. Report ionic strenght used in this set of experiments.
Line 174 and 176/177: Sodium or potassium? Please, add.
Sections 2.9 and 2.10: Have been the tests performed under static conditions? Please, comment.
Line 242: Correct, as suggested “...reached 100% for both conditions.”
Results and discussion section requires the description of catalyst concentration used in each set of experiments.
Section 3.2.1: The effect of molar ratio on the selectivity and conversion should be better discussed. The present discussion is very poor.
Section 3.2.2: It is hard to read this study. Green bases have been used as catalysts? Please, clarify. A representative scheme of the mechanism of reaction is required. It is more interesting to the readers. The effect of green bases on the selectivity and conversion should also be better discussed.
Section 3.2.3: The effect of organic solvents on the selectivity and conversion should also be better discussed.
Author Response
Comments and Suggestions for Authors
The present study describes the synthesis of EGCG palmitate derived from natural tea polyphenols. Stability and inhition studies have also been performed. Although the study is very interesting, the manuscript requires a deep revision to be published.
Title: What is EGCG? The authors should avoid here such abbreviation.
Response: Thanks for the suggestion. The full name (-)-epigallocatechin gallate has been added.
Abstract: The authors should describe the optimal experimental conditions and maximum yield.
Response: The optimal experimental condition has been described previously in 3.2.3 section: In conclusion, considering the reaction variables on effect of the reaction, the optimum reaction condition is set as follows: sodium acetate as the base, acetone as the solvent, and the molar ratio of palmitoyl chloride is 2.
The maximum yield has been added in 3.2.3: Under the optimum reaction condition, the yield of EGCG palmitate can reach 90.6%.
Lines 26-27, 46-47, and 57-58: Please, add references.
Response: The references had been added.
Lines 57-64: The authors should better explain the potential chemical catalysts used for such a purpose (homogeneous or heterogeneous catalysts). Among them, solid acid catalysts are more suitable from an industrial point of view due to easy recovery and reuse, as well as low wastes production during separation and purification steps. A deep revision in this topic is required. It is more interesting to the readers.
Response: Thanks for the suggestion. The description of homogeneous or heterogeneous catalysts has been added in Introduction.
Lines 66-70: How is this system different to other reports to merit publication? Has been a similar process reported in a previous study?? Please, report.
Response: The merits of this method are:
- This system used relatively non-toxic solvent (acetone) and solid base (sodium acetate). Comparison of the toxicity of some reagents and their harm to human health has been listed in Table 1.
- The isolation process is facile, since the excessive solid base is filtered and removed directly.
- The price of the regents used are significantly lower than the enzymes.
- In industries, traditional esterification is usually catalyzed by sulfuric acid. But the use of sulfuric acid easily causes secondary reactions, such as oxidation, sulfonation and carbonization. Besides, sulfuric acid etches the equipment and pollutes the environment. But solid bases used in this method don’t include these defects.
A similar process was reported in our previous study, but in that paper just the optimum condition for synthesis was described, and that paper focused on the antioxidant activity evaluation of EGCG palmitate (Ref. B. Liu, W. Yan, Lipophilization of EGCG and effects on antioxidant activities, Food Chem. 272 (2019) 663-669).
Line 73: Please, report EGCG source (plant species) used in this study. Information about the catalyst used in this set of experiments is strongly required.
Response: EGCG source (plant species) has been added: EGCG (from green tea, purity 92 %). Information about the catalyst used in this set of experiments has been added in 2.1 Materials section.
Lines 74 and 76: Please, correct scientific nomenclatures of enzyme sources (Bacillus licheniformis and Saccharomyces cerevisiae).
Response: The scientific nomenclatures of enzyme sources have been revised: α-amylase (from rhizobacteria Bacillus Licheniformis) and α-Glucosidase (from yeast Saccharomyces Cerevisiae).
Lines 81-91: The authors should better describe the general ester synthesis procedure (temperature control – immersed reaction in a water bath or jacketed reactor; magnetic or mechanical stirring and frequency; stirrer conformation). Please, describe. Again, information about the catalyst used in this study is strongly required.
Response: The detailed description of the general synthesis procedure has been revised in the revised version (2.2. Synthesis procedure). Information about the catalyst used has been added in the Materials section.
Lines 131-133: The authors should better explain what is PBS (phosphate buffer solution??). In addition, SODIUM or POTASSIUM? Ion strength used in this set of experiments.
Response: The full name of PBS is sodium phosphate buffer and is revised in the 2.1 Materials section: Sodium phosphate buffer (PBS) was purchased from Solarbio Co. (Beijing, China) and was prepared to make a concentration of 0.01 M aqueous solution (pH 7.2). Since the ion strength is not offered by the vendor, we can’t provide this information.
Lines 134-138: Explain how the temperature control has been performed in this set of experiments.
Response: The temperature was controlled by using a thermostat water bath.
Line 139 and others: I suggest replace in all text “long-term stability” by “storage stability”.
Response: Thanks for the suggestion. “Long-term stability” has been replaced by “storage stability”.
Line 153: “10 mM sodium phosphate buffer (pH 6.9)”. Report ionic strenght used in this set of experiments.
Response: Since the ion strength is not offered by the vendor, we can’t provide this information.
Line 174 and 176/177: Sodium or potassium? Please, add.
Response: It is sodium phosphate buffer. These have been revised.
Sections 2.9 and 2.10: Have been the tests performed under static conditions? Please, comment.
Response: Yes. The tests are single-factor experiments. Only one variable was investigated in the process.
Line 242: Correct, as suggested “...reached 100% for both conditions.”
Response: It has been revised into “...reached nearly 100% for both conditions.” And in corresponding Tables, 100% have been revised into nearly 100%.
Results and discussion section requires the description of catalyst concentration used in each set of experiments.
Response: The amount of the solid bases and triethylamine added is 2 molar ratio, respectively. But since the solid base is partly dissolved in the reaction, we can’t provide the concentration of these bases in each set of experiments.
Section 3.2.1: The effect of molar ratio on the selectivity and conversion should be better discussed. The present discussion is very poor.
Response: This part has been discussed in detail in 3.2.1. Effect of the molar ratio of palmitoyl chloride.
Section 3.2.2: It is hard to read this study. Green bases have been used as catalysts? Please, clarify. A representative scheme of the mechanism of reaction is required. It is more interesting to the readers. The effect of green bases on the selectivity and conversion should also be better discussed.
Response: Thanks for the comment. We searched for some documents and considered “green” is not appropriate to describe this method. We have revised the description of “green” into “novel” instead.
A representative scheme of the mechanism of the reaction has been added in the Graphical Abstract file.
The effect of green bases on the selectivity and conversion has been discussed in 3.2.2. Effect of base: By using 2 molar ratio of different bases, the products are still EGCG monopalmitate. The possible reason is that these bases don’t activate the reactivity of EGCG or palmitoyl chloride, but just react with the hydrogen chloride released in the reaction.
Section 3.2.3: The effect of organic solvents on the selectivity and conversion should also be better discussed.
Response: The effect of organic solvents on the selectivity and conversion has been discussed in 3.2.3. Effect of solvent: and the products were mono-esters.… In addition, since the solid base (sodium acetate) can only play the acid-binding role when it is dissolved in the solvent, its solubility in the solvent is a critical factor. With the polarity of the solvents decreasing, the solubility of the solid base decreased. For this reason, the solubility of
Reviewer 2 Report
L69 Finally, the antidiabetic activity of PEGCG was evaluated
State whether there are reported cases of the antidiabetic activity of PEGCG in the literature. Also, please state why you chose this activity.
L301 (Fig. 3.) The above results indicate that the PEGCG is significantly more stable than EGCG >>> The above results indicate that the PEGCG is significantly more stable as flavonoid than EGCG (PEGCG itself disappeared completely in 5 hours, and the representation, "PEGCG is stable" is incorrect.)
Reviewer 3 Report
I must say that the manuscript seems to be well arranged, with distinct topics and well discussed.
I suggest the authors to revise the manuscript according to the following comments before considering for publication.
In the abstract and in the introduction the authors define their synthetic method for EGCG palmitate as "green". I agree that the acetate base is better than pyridine, however, in what terms can the method be defined as £green”? Is the solvent green? Why? Is the reagent palmitoyl chloride green? Please justify this choice, also by adding (eventually) references.
Round 2
Reviewer 1 Report
The manuscript was corrected, as suggested. However, some minor corrections are still required before publication:
a) Abstract: The authors should briefly report the experimental conditions that maximize the reaction and yield.
b) Tables 2 and 3: The results should be represented as mean ± standard deviation of xxxx replications.
Reviewer 3 Report
The manuscript was corrected as suggested and it may now be published
Author Response
Comments and Suggestions for Authors
The manuscript was corrected as suggested and it may now be published.
Response: Thanks!